# Thiamine status of whitefish (*Coregonus maraena*) in the Baltic Sea

**Marc M. Hauber**[ID]*, **Oscar Nordahl, Vittoria Todisco, Emil Fridolfsson, Petter Tibblin, Samuel Hylander**[ID]

Centre for Ecology and Evolution in Microbial model Systems (EEMiS), Linnaeus University, Kalmar, Sweden

* marc.hauber@lnu.se

## Abstract

Many coregonine species have declined drastically across the Northern Hemisphere, including populations of *Coregonus maraena* (whitefish) in the Baltic Sea, and the mechanisms leading to these declines are not well investigated. An abrupt population crash occurred in the 1990s, coinciding with heavy declines in salmonid recruitment, also known as thiamine deficiency syndrome. Thiamine, i.e., vitamin B1, is an essential micronutrient needed for a functional metabolism. Offspring with thiamine deficiency have a high mortality posing significant negative impact on populations. Here, we aim to determine if whitefish, like other salmonids in the Baltic Sea, is affected by thiamine deficiency. Anadromous whitefish were therefore sampled during spawning in rivers of Southeastern Sweden, and we compared tissue concentrations and thiamine-dependent enzyme latencies to published thresholds. Further, we tested whether the variation in thiamine concentrations among individuals could be explained by physiological and morphological traits. Results showed that latency of thiamine-dependent enzymes along with egg thiamine concentrations suggest no evident thiamine deficiency. Concentrations were generally higher in the liver compared to muscle tissues. While females had lower liver thiamine concentrations compared to males, the opposite was found for muscle tissues, suggesting sex-specific patterns of allocation of the vitamin. Concentrations in eggs were positively related to the condition of the females and, similar to muscle and liver tissues, tended to be negatively related to standardized gill raker length. The latter is often used as a proxy for characterizing the feeding niche of coregonines. As has been observed in a number of other organisms (e.g., fish and molluscs), there was a reduction in thiamine concentration with length. Hence, the populations studied here showed no evidence of exhibiting thiamine deficiency. The variation in thiamine concentrations could largely be attributed to intrinsic physiological traits as well as traits associated with coregonine feeding niche.

**Data availability statement:** Data supporting this study are openly available from Dryad at: https://doi.org/10.5061/dryad.w0vt4b964.

**Funding:** This study was supported by VR (grant no.: 2019-04251) and FORMAS (2020/2020-01514) to S.H. Additionally, funding was provided by Linnaeus University and the Strategic Research Program Ecochange. There was no additional external funding received for this study. The funders had no role in study design, data collection and analysis, decision to publish, or preparation of the manuscript.

**Competing interests:** The authors have declared that no competing interests exist.

## Introduction

The fish taxon Coregoninae, a subfamily of Salmonidae, occurs in lentic and lotic habitats across the Northern hemisphere. For millennia, these fishes have been of great value to human kind whether it be nutritional, economical, or cultural [1–3]. Due to anthropogenic stressors and intense harvesting over the past century, many species of this taxon have suffered severe declines leading to population or even species extinctions [4–8].

One of these species is whitefish (*Coregonus maraena*; previously *C. lavaretus*). Once one of the most valuable target species for local fisheries in the Baltic Sea, whitefish is nowadays classified as endangered [9]. Particularly in the northern parts of the Baltic Sea, whitefish has been of great socioeconomic and cultural importance for centuries [10,11]. Hence, whitefish stocks were exploited to an increasing extent with landings often exceeding 4,000 t per year [2,12]. In the later half of the 20th century, whitefish stocks gradually declined. Stocking programs were established to support productivity but in the 1990s Baltic whitefish stocks collapsed and have remained at low abundances since [2,12,13].

Anthropogenic stressors such as overexploitation, habitat degradation, and hydropower plants hindering the completion of the life-cycle of anadromous populations have been suggested as main causes for the decline of Baltic whitefish [2,14]. Although these factors most likely impacted whitefish stocks, they might not fully explain the abrupt collapse in the 1990s. Firstly, most migratory barriers such as dams associated with hydropower plants or mills were constructed well before the 90s [15]. Secondly, whitefish populations have also drastically declined in areas where coastal commercial fisheries did not target them. In southern parts of Sweden, i.e., Småland and Blekinge, whitefish was only targeted by recreational fisheries [16]. Catches of such recreational fisheries did not distinctively increase in the years before the collapse [12]. Lastly, eutrophication and the associated habitat degradation in the Baltic Sea and its tributary rivers peaked in the 90s. Though regulations on nutrient input into riverine systems have improved the state of spawning grounds for river-spawning, i.e., anadromous, whitefish populations, spawning grounds of sea-spawning populations have seen, similarly to feeding grounds, a stabilisation in a highly eutrophic state [17,18].

Simultaneously to the collapse of whitefish in the 90s, other salmonid species living in the Baltic Sea, i.e., Atlantic salmon (*Salmo salar*) and sea trout (*Salmo trutta* morpha *trutta*), suffered from high mortality events in early life-stages, particularly in yolk-sack fry [19–21]. These large-scale mortality events were caused by a lack of vitamin B1, i.e., thiamine deficiency. Thiamine is an essential micronutrient crucial for the metabolic functionality of almost all organisms [22]. In an aquatic ecosystem, thiamine is only produced by primary producers such as bacteria and phytoplankton. All higher trophic levels need to acquire it by predation, symbiosis or diffusion [22,23]. Since thiamine deficiency has been observed in several salmonid species across the Northern Hemisphere for more than 50 years, a variety of causes, mainly addressing the quality of prey items, have been put forward [24]. However, a consensus on the mechanism behind the onset of thiamine deficiency has not been reached. This

makes it difficult to infer how prevalent thiamine deficiency is within an ecosystem and which species may be vulnerable to it. Inter- and intraspecific variation in thiamine levels further complicate our understanding. Whitefish presents an interesting study system in this context because of its relatedness to other salmonids and its variable niche utilization.

We set out to investigate whether Baltic whitefish populations are affected by thiamine deficiency as a potential causative agent for their collapse in the 90s. To do so, we analyzed the thiamine status of whitefish using the activity and saturation of thiamine-dependent enzymes as well as tissue-specific thiamine concentrations. We studied whether the measured variation relates to thiamine deficiency, or other factors such as demography, physiology or phenotype.

## Materials and methods

### Study system and sampling

Adult whitefish were sampled from three anadromous populations during spawning (November to December, 2020) in the rivers Lyckebyån, Skräbeån and Virån along the Swedish south-east coast (Fig 1A). In Virån (Kalmar County, n = 39), whitefish were caught using gill nets and cast nets. In Lyckebyån (Blekinge County, n = 20), dip nets were applied whereas electrofishing and gill nets were used in Skräbeån (Skåne County, n = 20). Additional female whitefish were caught in Virån (n = 13; 2020) and Skräbeån (n = 8; 2022) and stripped for egg samples only before being released back into the wild.

Each specimen collected for determining tissue thiamine concentrations was euthanized by percussion stunning and severing the gill arch to quickly bleed out. Afterwards, each specimen was sexed, weighted and its total length was noted (S1 File App). We then eviscerated the fish and weighted liver and the gastrointestinal tract separately. To investigate their thiamine status, we sampled a piece of dorsal muscle, liver, and eggs. As females were ovulating at time of sampling, their

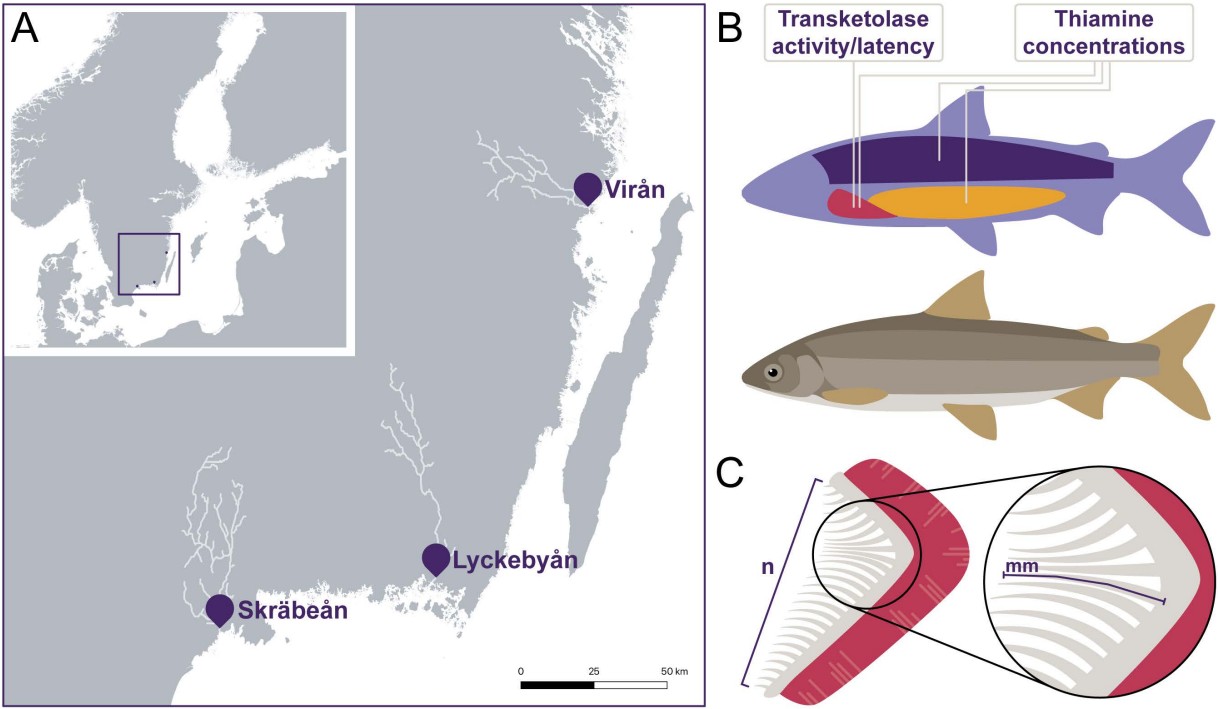

**Fig 1. Study location, organism, tissues and additional parameters. (A)** Map of Baltic Sea with zoom-in (purple square) to southeastern Swedish coast indicating the sampling locations in the three rivers Lyckebyån, Skräbeån, and Virån. **(B)** Illustrations of whitefish and the investigated tissues. Thiamine concentrations were measured in muscle, liver, and eggs. Transketolase activity and latency was measured in the liver. **(C)** Illustration of a gill arch with gill rakers. The number (n) of gill rakers was counted and the length of a specified gill raker was measured (zoom-in).

gonads were (partly) emptied. Hence, we could not sample eggs from all females and gonad weight could not be reliably determined. All tissue samples were transported on dry ice and stored at −80°C. The eviscerated whitefish were stored at −20°C for further processing (see section "Gill raker count and length" below). The sampling was conducted under ethical permit (5.2.18–482/14).

## Thiamine analysis

Thiamine was analyzed in muscle, liver, and eggs according to [25] with slight modifications. For a detailed description of the process see Todisco et al. [26]. While liver and egg samples were analyzed as described in Todisco et al. [26], muscle tissue (~ 1 g) was analyzed in half the stated volume of 2% and 10% trichloroacetic acid as thiamine concentrations in muscle tissue were comparatively low. In short, a subsample of tissue was homogenized and boiled in diluted trichloro-acetic acid. After centrifugation, the supernatant was washed with a mixture of ethyl acetate and hexane. Lastly, we added the dye $K_3Fe(CN)_6$ and filtered the mixture before it was injected into a Hitachi Chromaster HPLC system to measure fluorescence. We quantified three thiamine vitamers: free thiamine (TF), thiamine monophosphate (TMP), and thiamine diphosphate (TDP). Their concentrations were normalized per g wet weight and summed to total thiamine concentration (Ttot). Vitamer ratios were studied in muscle and liver samples but since whitefish eggs partly thawed during sample preparation, shifts in vitamer ratios may have occurred. Hence, we refrained from studying vitamer ratios in eggs.

## Transketolase activity

The enzymatic functionality of transketolase is dependent on TDP binding to it. In the matrix of thiamine deficient organisms, transketolase lacking TDP may be present which renders these enzymes inactive. Measurements for the presence of such unsaturated transketolase, i.e., latency, can be used as an indicator for the thiamine status of an organism. Commonly, latencies above 20% are categorized as thiamine deficient [27].

We measured liver transketolase activity and latency using the BCA Protein Assay Kit (ab102536, Abcam) and Transketolase Activity Assay Kit (ab273310, Abcam) as described in Hauber et al. [28] including modifications from Jones et al. [27]. In short, liver tissues were homogenized in Tris Buffer, centrifuged and filtered. Samples were diluted to reach protein concentrations between 0.2–0.4 µg/µl. Following the manufacturer's protocol, assays for measurement of the basal transketolase activity were prepared. We added assays infused with TDP to measure the saturated transketolase activity. Standards, substrate control, sample background controls, and positive control were prepared following the manufacturer's protocol. Using the microplate reader (FLUOstar Omega, BMG Labtech) we measured fluorescence every minute for 60 min. Following the manufacturer's protocol, we calculated an average basal and stimulated transketolase activity for each specimen. The latency was calculated by dividing the basal activity by stimulated activity, subtracting it from 1 and multiplying it by 100.

Three of the samples were run on a separate plate and show much lower, only a quarter as high, transketolase activities. We believe this difference between plates is not of biological origin but caused by a misstep during manufacturing or protocol preparation. Hence, we excluded these three samples from the statistical analysis.

## Egg coloration, weight and water content

We measured egg parameters to assess whether thiamine concentrations relate to egg quality to get a better understanding of the variation in egg thiamine concentrations.

Egg pigmentation, particularly the presence of carotenoids, was measured as an additional indicator for egg quality. Around 1 g of thawed eggs were spread out in an aluminium cup and photographed from above in a standardized setup under consistent overhead lighting in a windowless room (Canon EOS 700D; ISO 200, shutter speed 1/25 s). In Photoshop (version 21.1.0 © 2020 Adobe), the area covered by eggs in each picture was selected and we measured

an average value in the CIE (Commission Internationale de l'Eclairage) L*a*b* color space. This standardized, uniform and device independent color space separates luminescence (L*) from two measures of color intensity (a* and b*). Both channel a* and b* can be used to indirectly measure the presence of pigments [29,30]. The a* channel correlates with red carotenoids such as astaxanthin, the b* channel correlates with yellow carotenoids such as lutein [31,32].

After the eggs were photographed, they were dried at 60°C for 48 h and weighed again to calculate their water content. Lastly, the number of eggs was counted to calculate weight per egg.

### Gill raker count and length

Even within the same environment, whitefish can display variation in phenotype and niche utilization, resulting in divergent ecotypes. These ecotypes differ in functional morphological traits that correlate with habitat use and foraging strategies. In particular, the count and size of protrusions from the gill arches, i.e., gill rakers, may indicate the usage of different trophic niches [33,34].

To study the gill raker count and length, we thawed the eviscerated fish and removed the first left gill arch. We took standardized pictures (Canon EOS 700D) of the gill arch and counted the number of gill rakers (Fig 1C, S1 File App). Lastly, we measured the length of the first (from top) gill raker on the lower gill arch (Fig 1C) using Fiji (an ImageJ distribution, version 1.53c) [35]. We could not determine the number of gill rakers for one individual from Lyckebyån as the gill arch had been damaged.

### Statistical analysis

To test the effect of physiological as well as morphological traits on thiamine concentrations of somatic tissues, i.e., muscle and liver, we fitted a linear model with liver and muscle thiamine concentrations as the response variable. Tissue type, population, sex, length, condition, standardized gill raker length and gill raker count were included as fixed effects, assuming a Gaussian error distribution. Interaction terms were added for tissue type with population and sex. Lastly, we included a unique identifier for each fish as a random intercept using the lme4 package [36]. We calculated the condition, used as a proxy for health, by fitting total weight against length, both log-transformed, and extracting the residuals. We standardized gill raker length in the same way.

To investigate whether maternal effects or egg quality relate to thiamine concentrations in reproductive tissues, i.e., eggs, we fitted two linear models with egg thiamine concentrations as response variable. We ran two models, since some egg samples were from fish that were released back into the wild and therefore we were lacking maternal data. Maternal effects on egg thiamine concentrations were tested by including length, condition, standardized gill raker length, and muscle and liver thiamine concentrations as fixed effects. To test whether egg quality relates to egg thiamine concentrations we included channel a*, channel b*, egg water content, and weight per egg as fixed effects.

Lastly, transketolase activity and latency were used as response variables in two separate models that included length, condition, sampling location, and liver thiamine concentration as fixed effects. Length was scaled in all models. We evaluated model fit using diagnostic plots of residual distribution with the DHARMa package [37]. P-values were computed based on robust covariance matrix estimation, i.e., Wald tests. All statistical analyses were performed using R (version 4.4.2) [38].

## Results

### Thiamine in somatic tissues

Liver thiamine concentrations (mean: 18.5±4.38 nmol/g) were up to ten times higher than those measured in muscle tissue (mean: 2.43±0.36 nmol/g; Fig 2). Vitamer ratios between liver and muscle were rather similar with TDP being most prominent, followed by TMP and TF (S2 File App). Liver and muscle thiamine concentrations showed significant

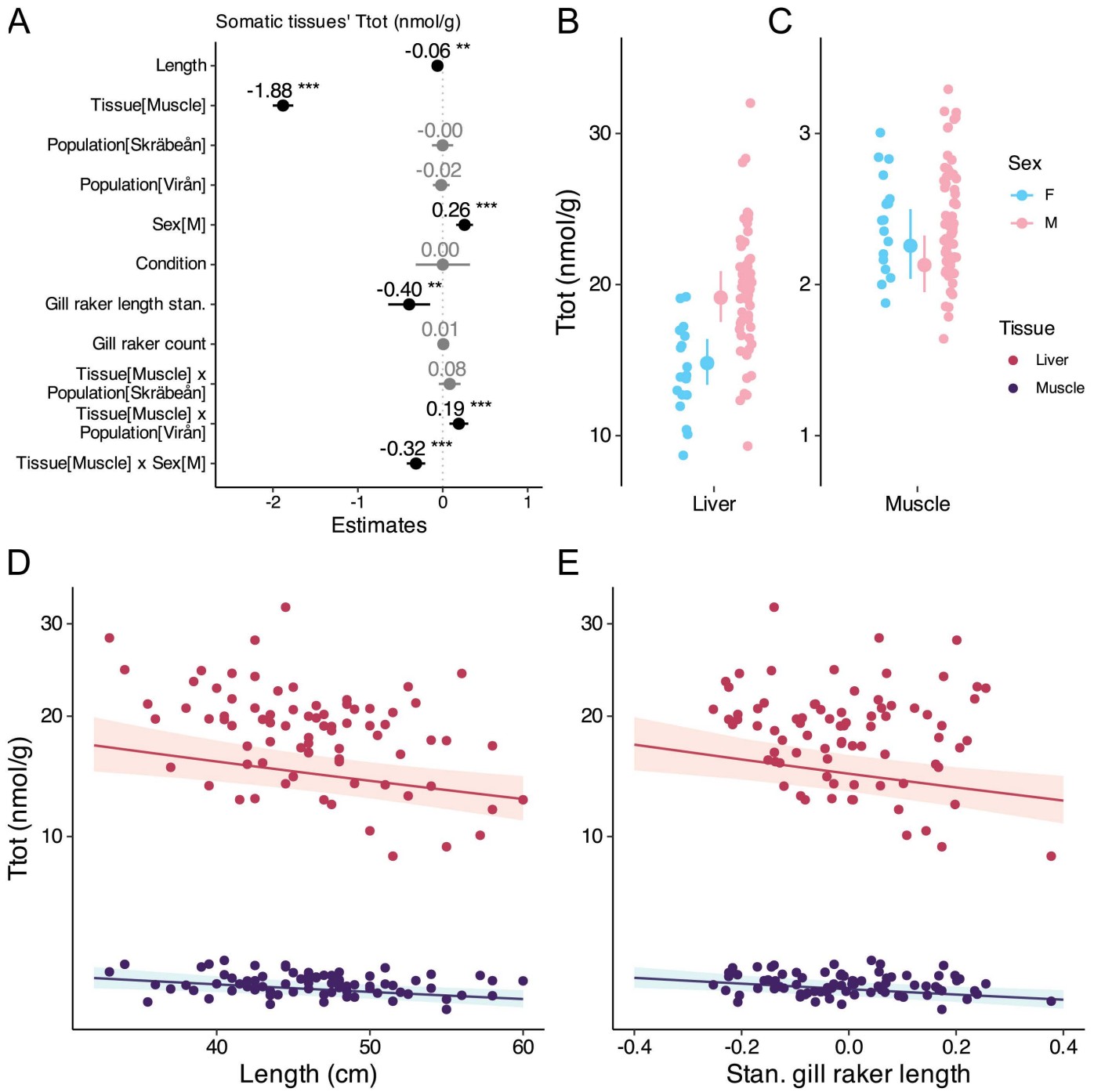

**Fig 2. Model output investigating Ttot (nmol/g) in muscle and liver tissues. (A)** Statistically significant effects are indicated by an asterisks where *** p < 0.001. **(B, C)** Predictions of the model with 95% confidence intervals and raw data show sex specific differences in thiamine concentrations in both tissues, **(D, E)** and an overall decline in somatic thiamine concentrations with length and standardized gill raker length. **(D, E)** Please note the y axis was square root transformed.

interactions with sex and population (Fig 2A, S3 File App). Whereas thiamine concentrations in muscle tissue were significantly higher for female whitefish, their liver thiamine concentrations were significantly lower compared to males (Fig 2B and 2C). While liver thiamine concentrations were similar between the populations, muscle thiamine concentrations were significantly higher in Virån compared to Lyckebyån (Fig 2A). Overall, somatic thiamine concentrations declined with length and the standardized gill raker length (Fig 2D and 2E, S3 File App). Somatic thiamine concentrations showed no significant relationship with gill raker count and condition (Fig 2A, S3 File App).

### Thiamine in eggs

Egg thiamine concentrations averaged around 12 ± 3.86 nmol/g and ranged from 6.35 to 21.35 nmol/g. They significantly increased with the female's condition and showed a marginally non-significant decline with the female's standardized gill raker length (Fig 3, S3 File App). Other maternal parameters, length and somatic thiamine concentrations, did not show

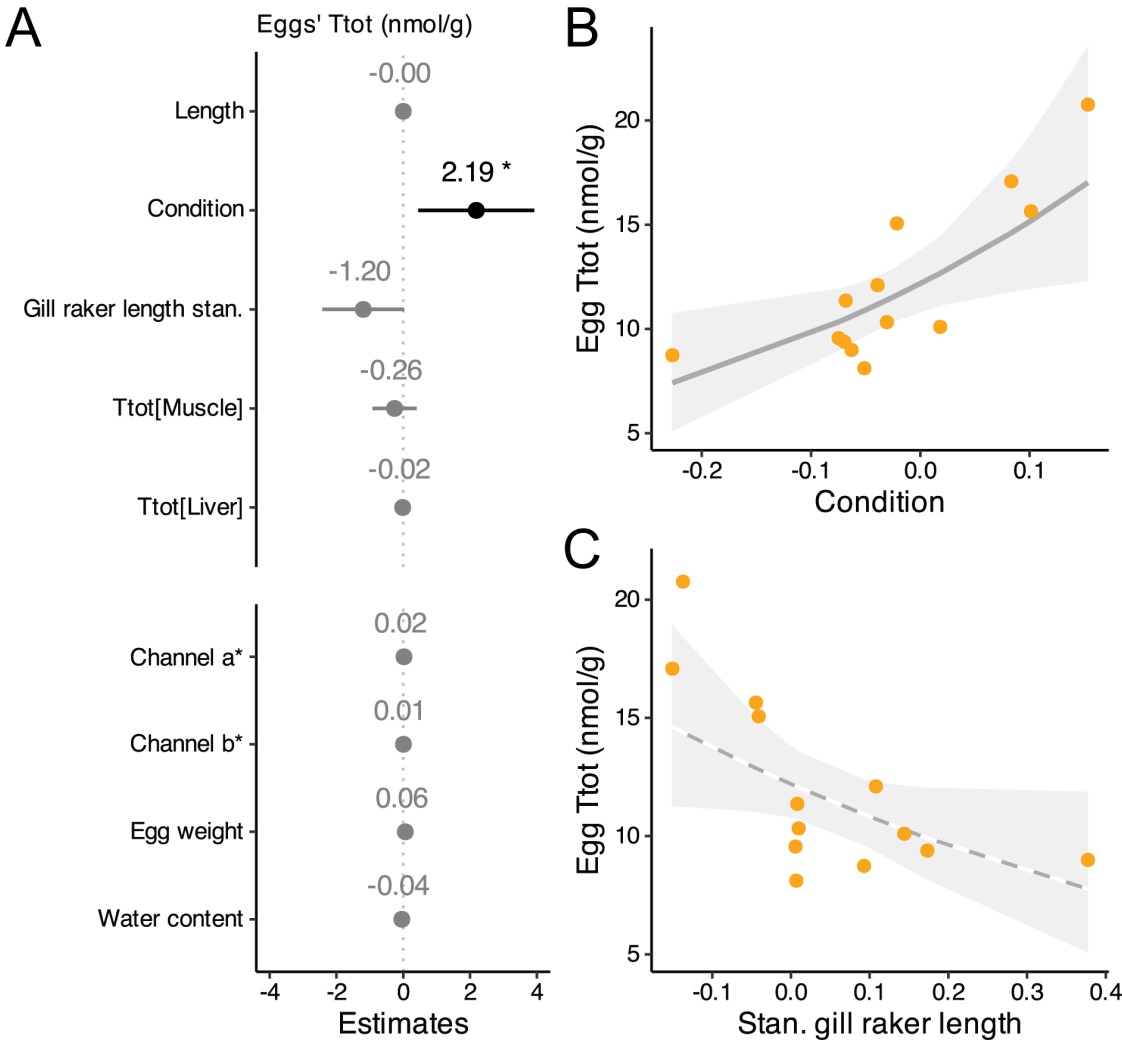

**Fig 3. Model outputs investigating Ttot (nmol/g) in eggs. (A)** Statistically significant effects are indicated by an asterisk where *** p < 0.001. **(B)** Predictions of the models with 95% confidence intervals and raw data show that egg thiamine concentrations increase with female condition, and **(C)** decrease, marginally non-significant (p = 0.054), with standardized gill raker length.

any relationship with egg thiamine concentrations (Fig 3A, S3 File App). None of the measured parameters on egg quality including pigmentation (channel a*, channel b*), egg weight, and water content showed any relationship with egg thiamine concentration (Fig 3A, S3 File App).

## Transketolase

Transketolase activity averaged around 73.9 ± 28.2 µUnits/µg and varied from 22.63 to 121.64 µUnits/µg (Fig 4A). None of the investigated variables, including population, length, condition, and thiamine concentrations in liver tissue, showed a significant relationship with transketolase activity or latency (S4 File App). Transketolase latency for all females averaged at 1.32 ± 3.33% and ranged from −5.33 to 7.57% (Fig 4B). Latency values did not exceed the 20% threshold in any of the investigated individuals, indicating that none of the sampled fish met the criteria for thiamine deficiency.

## Discussion

To determine whether whitefish populations in the Baltic Sea are suffering from thiamine deficiency, we investigated thiamine concentrations in different tissues of whitefish and measured the activity and latency of transketolase. We found that (i) thiamine concentrations of somatic tissues differ between the sexes and decline with length; (ii) thiamine concentrations of eggs increase with their mother's condition; (iii) thiamine concentrations decrease with standardized gill raker length; (iv) and transketolase latencies clustered around 0%.

We assessed the thiamine status of whitefish by comparative investigations of the egg thiamine concentration and the latency of transketolase. Though the measured somatic thiamine concentrations contribute to our broader understanding of thiamine dynamics within this species and in fish in general, thiamine concentrations in these tissues showed different levels than other salmonid species [26,39]. Although such interspecific comparisons of somatic tissues concentrations do not allow direct inference on whether whitefish suffered from thiamine deficiency, it was notable that thiamine concentrations of eggs were comparable to those of healthy specimens of other salmonids [see, e.g., 26,39,40]. For salmonids such as Atlantic salmon, Lake trout (*Salvelinus namaycush*), and Coho salmon (*Oncorhynchus kisutch*), the thresholds of egg thiamine concentrations that are expected to have a lethal effect on the offspring is known, but for the genus Coregonus

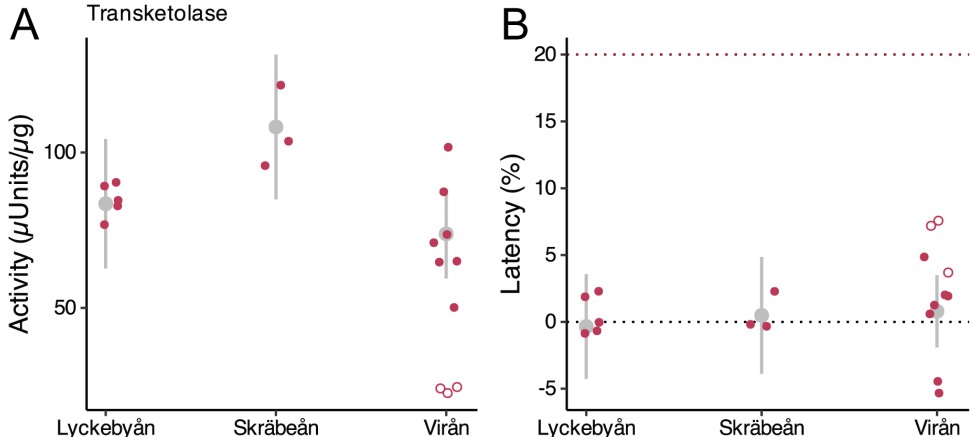

**Fig 4. Predictions of models and raw data on transketolase activity (A) and latency (B).** Points outlined in red were run separately and were excluded from statistical analysis due to their lower measured activity (see methods section). Predictions of model and raw data on transketolase latency show values averaging around zero. Threshold indicating thiamine deficiency, at 20%, is represented by a red dotted line. Negative latency values are not biologically meaningful but are presented to indicate range of measurement inaccuracy.

this knowledge is lacking. Riley et al. [41] studied thiamine in lake whitefish (*C. clupeaformis*) and found concentrations in eggs (mean: 8.78 nmol/g, variation: < 1–29 nmol/g) comparable to slightly lower than our study. To assess the thiamine status of lake whitefish, Riley et al. [41] applied a conservative approach in which the proportion of samples below putative thresholds for lethal (1 nmol/g) and sublethal (4 nmol/g) effects were calculated. These cut-offs were based on thresholds found in the previously mentioned salmonid species [40,42–45]. When applying these thresholds to our data, none of the sampled whitefish eggs would be at risk of (sub-)lethal effects due to a lack of thiamine. Secondly, the enzyme transketolase did not show any latency, i.e., all enzymes appear to be saturated with TDP. This suggests that enough thiamine was present in the matrix of somatic tissues to ensure a functioning basic metabolism in adult fish. Combining these observations, we argue that whitefish in the Baltic Sea does not seem to suffer from thiamine deficiency. However, a caveat is that the frequency of deficiency in other species varies among years [20] and future studies should collect these variables over time to ensure that deficiency does not develop in particular years.

Comparative approaches, such as contrasting vulnerability to thiamine deficiency among species occupying different ecological niches within the same ecosystem, can help to disentangle potential underlying mechanisms and drivers of the deficiency. In the Baltic Sea, thiamine deficiency has been studied in a variety of fish species. While it is uncontested that Atlantic salmon is affected by thiamine deficiency, limited or contrasting information is present for other species. Engelhardt et al. [46] found Atlantic cod (*Gadus morhua*) to be affected by thiamine deficiency, but a recent study found contrastingly no evidence for thiamine deficiency in these stocks [28]. Similar to whitefish, European perch (*Perca fluviatilis*) and Atlantic herring (*Clupea harengus*) do not appear to suffer from thiamine deficiency in the Baltic Sea [47,48]. The diet of these three species is rather varied and encompasses zooplankton, benthic macroinvertebrates and partly other small fish [2,49–51]. Most of these diet items are of lower trophic level and therefore can be expected to be of higher thiamine content compared to a strictly piscivorous diet [23]. Especially crustaceans appear to be rich in thiamine compared to fish [52–54]. It should be noted that Atlantic cod also partly feed on benthic macroinvertebrates [55]. Based on these observations, one could hypothesize that especially fish species feeding on higher trophic levels, i.e., mainly piscivorous diets, may be most vulnerable to thiamine deficiency. This would overlap with our current knowledge in which we observe thiamine deficiency mainly in salmonids of similar life history who have fed on or have shifted towards feeding on a fish dominated diet [26,54,56]. Many of these species show long migration and starvation periods before spawning, potentially indicating that such life history traits additionally increase the vulnerability to thiamine deficiency. However, further investigations into other species of varying life histories are needed to critically assess this hypothesis.

Tissue-specific thiamine concentrations also related to other variables such as length, sex, condition and standardized gill raker length. The decline of thiamine concentrations in somatic tissues may be explained by the allometric, not proportional increase of metabolic rate with growth [57,58]. Hence, the demand for thiamine should scale similarly and decrease with size which has been observed in several studies [28,46,59,60]. Sex appears to have contrasting effects on muscle and liver thiamine concentrations. Female whitefish have higher concentrations in muscle tissues compared to males, a pattern previously also described for spawning Atlantic salmon [26]. Liver concentrations show an opposite pattern in which male whitefish have considerably higher liver concentrations than their female counterparts. Thiamine is almost exclusively provided to the offspring by the female. It could thus be hypothesized that this pattern arises from sex-specific differences in the timing and demand of thiamine allocation to gonadal tissues [61]. Among females, egg thiamine concentrations further showed a positive correlation with condition, used as a proxy of health. Since we could not determine gonad weight for female fish, the condition was calculated on total and not somatic weight. Hence, we cannot disentangle whether this relationship is due to female condition, reproductive output or spawning stage.

Lastly, thiamine concentrations in all tissues showed negative relationships with the standardized gill raker length. Typically, morphs with more and longer filaments specialize in feeding on smaller prey such as zooplankton in pelagic habitats, whereas benthivorous morphs tend to have fewer and shorter filaments [33,34]. Based on this pattern, our results suggest that foraging at lower trophic levels may not be associated with the highest thiamine levels, contrary to

our expectations. However, it is worth noting that the on average 26 gill rakers with little variation (S1 File App) counted for each specimen would be categorized as sparsely-rakered ecotypes and hence, true planktivores may not have been present in our study. Although our approach does not allow for a direct link between thiamine concentrations and niche utilisation, the results indicate that even slight variation in morphological traits may lead to differences in thiamine availability and/or demand.

In conclusion, our results suggest that whitefish does not suffer from thiamine deficiency, furthering our understanding of the symptom's prevalence across systems. The variability in thiamine observed here can be attributed to intrinsic physiological as well as morphological traits. By acknowledging the critical status of whitefish, we hope to encourage future research to disentangle the factors leading to its decline, and the eventual management to recover the abundance of this important species.

## Supporting information

**S1 File. App: Overview of sampled whitefish separated by population and sex.** Mean values and standard deviation is presented for total length, weight, gill raker length, and gill raker count.
(PDF)

**S2 File. App: Relative vitamer ratios in muscle and liver tissue of whitefish separated by sex.** Vitamers: free thiamine (TF), thiamine monophosphate (TMP), thiamine diphosphate (TDP). N-values are given in white at the bottom of each bar.
(PNG)

**S3 File. App: Summary of model outputs on thiamine concentrations in somatic tissues (muscle and liver) and eggs.**
(XLSX)

**S4 File. App: Summary of model outputs on transketolase activity and latency.**
(XLSX)

## Acknowledgments

We would like to thank Linus Lariander from Länsstyrelsen Skåne, Martin Stålhammar from Länsstyrelsen Blekinge, Tobias Borger from Länsstyrelsen Kalmar, Henrik Flink, Marcus Hall, and the local fishers for their support in planning and performing the sampling. We would also like to thank the landowners for granting us to perform the sampling on their land. We would like to thank Elin Kärvegård for her support during the laboratory analysis.

## Author contributions

**Conceptualization:** Marc M. Hauber, Oscar Nordahl, Vittoria Todisco, Emil Fridolfsson, Petter Tibblin, Samuel Hylander.

**Data curation:** Marc M. Hauber.

**Formal analysis:** Marc M. Hauber, Oscar Nordahl.

**Funding acquisition:** Petter Tibblin, Samuel Hylander.

**Investigation:** Marc M. Hauber, Oscar Nordahl, Vittoria Todisco, Emil Fridolfsson.

**Methodology:** Marc M. Hauber, Emil Fridolfsson.

**Project administration:** Marc M. Hauber, Oscar Nordahl, Petter Tibblin, Samuel Hylander.

**Resources:** Marc M. Hauber, Emil Fridolfsson, Samuel Hylander.

**Software:** Marc M. Hauber, Oscar Nordahl.

**Supervision:** Oscar Nordahl, Petter Tibblin, Samuel Hylander.

**Validation:** Marc M. Hauber, Oscar Nordahl, Emil Fridolfsson.

**Visualization:** Marc M. Hauber.

**Writing – original draft:** Marc M. Hauber.

**Writing – review & editing:** Oscar Nordahl, Vittoria Todisco, Emil Fridolfsson, Petter Tibblin, Samuel Hylander.

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
