## [Decision Letter · Decision Letter 0]

21 Jan 2026

*Coregonus maraena*

Dear Dr. Marc Maximilian Hauber,

Thank you for submitting your manuscript to PLOS ONE. After careful consideration, we feel that it has merit but does not fully meet PLOS ONE’s publication criteria as it currently stands. Therefore, we invite you to submit a revised version of the manuscript that addresses the points raised during the review process.

We look forward to receiving your revised manuscript.

Kind regards,

Amel Mohamed El Asely

Academic Editor

PLOS One

Journal Requirements:

“This study was supported by VR (grant no.: 2019-04251) and FORMAS (FR-2020/ 0008) to S.H. Additionally, funding was provided by the Strategic Research Program Ecochange.”

“This study was supported by VR (grant no.: 2019-04251) and FORMAS (FR-2020/ 0008) to S.H. Additionally, funding was provided by the Strategic Research Program Ecochange.”

4. In the online submission form, you indicated that your data will be submitted to a repository upon acceptance.  We strongly recommend all authors deposit their data before acceptance, as the process can be lengthy and hold up publication timelines. Please note that, though access restrictions are acceptable now, your entire minimal  dataset will need to be made freely accessible if your manuscript is accepted for publication. This policy applies to all data except where public deposition would breach compliance with the protocol approved by your research ethics board. If you are unable to adhere to our open data policy, please kindly revise your statement to explain your reasoning and we will seek the editor's input on an exemption.

5. Please be informed that funding information should not appear in the Acknowledgments section or other areas of your manuscript. We will only publish funding information present in the Funding Statement section of the online submission form. Please remove any funding-related text from the manuscript.

6. Please include captions for your Supporting Information files at the end of your manuscript, and update any in-text citations to match accordingly. Please see our Supporting Information guidelines for more information: http://journals.plos.org/plosone/s/supporting-information .

7. We note that Figure 1 in your submission contain [map/satellite] images which may be copyrighted. All PLOS content is published under the Creative Commons Attribution License (CC BY 4.0), which means that the manuscript, images, and Supporting Information files will be freely available online, and any third party is permitted to access, download, copy, distribute, and use these materials in any way, even commercially, with proper attribution. For these reasons, we cannot publish previously copyrighted maps or satellite images created using proprietary data, such as Google software (Google Maps, Street View, and Earth). For more information, see our copyright guidelines: http://journals.plos.org/plosone/s/licenses-and-copyright.

Reviewers' comments:

Reviewer's Responses to Questions

**Comments to the Author**

1. Is the manuscript technically sound, and do the data support the conclusions?

Reviewer #1: Yes

Reviewer #2: Partly

2. Has the statistical analysis been performed appropriately and rigorously?

Reviewer #1: Yes

Reviewer #2: Yes

3. Have the authors made all data underlying the findings in their manuscript fully available?

Reviewer #1: No

Reviewer #2: Yes

4. Is the manuscript presented in an intelligible fashion and written in standard English?

Reviewer #1: Yes

Reviewer #2: Yes

Reviewer #1: I have reviewed the manuscript Thiamine Status of Whitefish (*Coregonus maraena* ) in the Baltic Sea. The authors have investigated thiamine in different tissues and egg during the spawning run of three anadromous populations in southern Sweden and conclude that thiamine deficiency is, probably, not a problem for Baltic whitefish. It is well-written and clearly structured. I enjoyed reading it.

As there is a general need to better map and understand the role of thiamine, this study provides useful information for whitefish and the prevalence of thiamine deficiency in the Baltic ecosystem. Some results were as expected, e.g., regarding sex-specific differences and relationship between female condition and egg concentrations, while others appear counter-intuitive, e.g., concentrations decrease with standardized gill raker length. For several variables no relationships could be detected.

Whitefish are known to be plastic and the study of ecomorphs have a long tradition. There appears consensus that increasing gill raker length indicates a more planktivorous diet, but could the actual length (and number) be a reason for the counter-intuitive results? Other studies have termed ~40 rakers as “medium-rakered” (Sarvala et al. 2024), while the raker count in this study is ~26 (with little variation). Perhaps the lack of effect and contrasting result to hypothesis be due to not having “true” planktivores in the samples? Perhaps it is worth mentioning in the discussion (L319-326)?

Related to the low variation in raker counts, but larger range/variation in length, it makes sense that gill raker length rather than counts came out as a significant result. Together with lack of support for several variables this raised an overall question for me, given the sample sizes and the ratio of samples to predictor variables – how likely were the authors to detect such effects? I do not suggest any ad-hoc power analyses, and although uncertainty for parameter estimates is given, I wonder if the authors have any thoughts on this that should be included in the manuscript? Was there for example differences in egg coloration, weight and water content across populations? Or could a relationship not be detected due to not having “enough” variation in the samples rather than the number of samples?

I want to also say that the uncertainty associated with using the 20% threshold extrapolated from another species is well described and motivated, as this could otherwise have been a source of critique.

I have also one detailed comment. On line 58, the authors give causes of whitefish decline in the Baltic, but the reference (15) appears unrelated. Perhaps another reference is more suitable, or should the sentence be rephrased?

References

Sarvala, J., Helminen, H., Karjalainen, J., Marjomäki, T.J., Forsman, T. & Anttila, L. (2024). Long-term decline of whitefish (Coregonus lavaretus) population in the boreal lake Pyhäjärvi, southwest Finland, relative to simultaneous abiotic and biotic changes. International Journal of Limnology, 60, 16. https://doi.org/10.1051/limn/2024009

Reviewer #2: First of all, I would like to thank the authors for this study “Thiamine Status of Whitefish (Coregonus maraena) in the Baltic Sea” and for the amount of work they performed throughout it. However, when going through the manuscript, I have some concerns that need to be addressed before considering this manuscript for publication.

Comments:

1. The abstract is clear, but a brief sentence summarizing the main findings would improve readability.Line21-23: please clarify the biological and nutritional relevance of thiamine in fish, particularly in relation to metabolism and reproduction.

2. Line24-26: the abstract would benefit from a brief mention of dietary factors related to thiamine deficiency to provide adequate nutritional context.

3. Line31-34: results should be presented depending on nutritionally relevant patterns as tissue specific and sex related differences.

4. The Introduction is well written, logically structured, and provides a strong ecological and historical context for the decline of Baltic whitefish.

5. Some paragraphs (particularly lines 58–71) are dense and could be slightly condensed without loss of meaning. This would improve readability.

6. Using the species name (Coregonus maraena) more often instead of the general term “whitefish” would improve clarity.

7. The methodology is sound and requires only minor clarification. Using the scientific name (Coregonus maraena) more often instead of “whitefish” would improve clarity.

8. The Results are clear and well supported by the analyses.

9. The Discussion is well organized and clearly interprets the results. The authors effectively place their findings in the context of other salmonid species and provide explanations for observed patterns.

10. It would help to start the Discussion with a short summary of the main results.

11. References and figures are well organized and relevant. No major changes needed.

**Do you want your identity to be public for this peer review?** For information about this choice, including consent withdrawal, please see our Privacy Policy

Reviewer #1: No

Reviewer #2: No

---

## [Author Response · Author response to Decision Letter 1]

4 Feb 2026

Reviewers' comments:

Reviewer #1: I have reviewed the manuscript Thiamine Status of Whitefish (Coregonus maraena) in the Baltic Sea. The authors have investigated thiamine in different tissues and egg during the spawning run of three anadromous populations in southern Sweden and conclude that thiamine deficiency is, probably, not a problem for Baltic whitefish. It is well-written and clearly structured. I enjoyed reading it.

Re. 10: Thank you for this positive review of our study. We are happy to hear that our writing was not only understandable but even enjoyable!

As there is a general need to better map and understand the role of thiamine, this study provides useful information for whitefish and the prevalence of thiamine deficiency in the Baltic ecosystem. Some results were as expected, e.g., regarding sex-specific differences and relationship between female condition and egg concentrations, while others appear counter-intuitive, e.g., concentrations decrease with standardized gill raker length. For several variables no relationships could be detected.

Whitefish are known to be plastic and the study of ecomorphs have a long tradition. There appears consensus that increasing gill raker length indicates a more planktivorous diet, but could the actual length (and number) be a reason for the counter-intuitive results? Other studies have termed ~40 rakers as “medium-rakered” (Sarvala et al. 2024), while the raker count in this study is ~26 (with little variation). Perhaps the lack of effect and contrasting result to hypothesis be due to not having “true” planktivores in the samples? Perhaps it is worth mentioning in the discussion (L319-326)?

Re. 11: Thank you for this comment. We have added a sentence mentioning the actual gill raker count and how they may indicate a lack of “true” planktivores in our study. This is a great addition to the discussion, as the gill raker count had not really been addressed before (L. 319-321).

Related to the low variation in raker counts, but larger range/variation in length, it makes sense that gill raker length rather than counts came out as a significant result. Together with lack of support for several variables this raised an overall question for me, given the sample sizes and the ratio of samples to predictor variables – how likely were the authors to detect such effects? I do not suggest any ad-hoc power analyses, and although uncertainty for parameter estimates is given, I wonder if the authors have any thoughts on this that should be included in the manuscript? Was there for example differences in egg coloration, weight and water content across populations? Or could a relationship not be detected due to not having “enough” variation in the samples rather than the number of samples?

Re. 12: This is a great point! We had decided to investigate many of the parameters within this study already during the study design. Especially the variability of whitefish ecotypes was of interest to us, since different diet preferences may affect the thiamine status in different ways. To our knowledge, whitefish ecotypes had not been investigated in these populations before, so we could not infer before the data collection how much variation we would observe. Egg coloration, weight and water content were included in the study, since we observed differences in egg coloration and consistency in the stripped egg samples during the sampling campaign. Hence, for these egg variables, we’d argue that enough variation was present to detect potential relationships with thiamine concentrations. However, when it comes to e.g. the gill raker count, the lack of variation in our study system hampered our investigation (now discussed L. 319-321). If repeated, it would be interesting to investigate thiamine concentrations in a system where several whitefish ecotypes have been described. However, the main aim of this study was to understand the thiamine status of whitefish in the Baltic Sea specifically and for this investigation the number and variation in samples was adequate. We believe that the addition made to the discussion regarding gill raker counts (Re. 11) adds to this topic in the manuscript and has improved the quality of the text. We are open to adding more information if the reviewer and/or editor believes it would benefit the manuscript.

I want to also say that the uncertainty associated with using the 20% threshold extrapolated from another species is well described and motivated, as this could otherwise have been a source of critique.

Re. 13: Thank you, we are glad to hear!

I have also one detailed comment. On line 58, the authors give causes of whitefish decline in the Baltic, but the reference (15) appears unrelated. Perhaps another reference is more suitable, or should the sentence be rephrased?

Re. 14: We have replaced the reference with more suitable ones. Thank you for finding this mistake (L. 49).

References

Sarvala, J., Helminen, H., Karjalainen, J., Marjomäki, T.J., Forsman, T. & Anttila, L. (2024). Long-term decline of whitefish (Coregonus lavaretus) population in the boreal lake Pyhäjärvi, southwest Finland, relative to simultaneous abiotic and biotic changes. International Journal of Limnology, 60, 16. https://doi.org/10.1051/limn/2024009

Reviewer #2: First of all, I would like to thank the authors for this study “Thiamine Status of Whitefish (Coregonus maraena) in the Baltic Sea” and for the amount of work they performed throughout it. However, when going through the manuscript, I have some concerns that need to be addressed before considering this manuscript for publication.

Re. 15: Thank you for reviewing our article and seeing the amount of work we have put into it.

Comments:

1. The abstract is clear, but a brief sentence summarizing the main findings would improve readability. Line21-23: please clarify the biological and nutritional relevance of thiamine in fish, particularly in relation to metabolism and reproduction.

Re. 16: Great point, we added a sentence about this (L. 15-16).

2. Line24-26: the abstract would benefit from a brief mention of dietary factors related to thiamine deficiency to provide adequate nutritional context.

Re. 17: We would exceed the maximum word count for an abstract with this addition. We have added some more context about this to the introduction (L. 68).

3. Line31-34: results should be presented depending on nutritionally relevant patterns as tissue specific and sex related differences.

Re. 18: We do not understand this comment. We have revised the abstract according to the previous comment but please let us know if the abstract needs more revision.

4. The Introduction is well written, logically structured, and provides a strong ecological and historical context for the decline of Baltic whitefish.

Re. 19: Thank you, we are glad to hear!

5. Some paragraphs (particularly lines 58–71) are dense and could be slightly condensed without loss of meaning. This would improve readability.

Re. 20: We have tried to condense this section. (L. 47-68).

6. Using the species name (Coregonus maraena) more often instead of the general term “whitefish” would improve clarity.

Re. 21: We understand the reviewer’s concern. However, the species C. maraena is most commonly referred to as whitefish and to keep the manuscript cohesive and more easily approachable for non-academic readers, we prefer to keep it the way it is.

7. The methodology is sound and requires only minor clarification. Using the scientific name (Coregonus maraena) more often instead of “whitefish” would improve clarity.

Re. 22: See Re. 21.

8. The Results are clear and well supported by the analyses.

9. The Discussion is well organized and clearly interprets the results. The authors effectively place their findings in the context of other salmonid species and provide explanations for observed patterns.

Re. 23: Thank you very much for this positive assessment.

10. It would help to start the Discussion with a short summary of the main results.

Re 24: We are not sure what the reviewer means here. We believe that the first paragraph of the discussion already adequately summarizes the study and its main results (L. 247-252).

11. References and figures are well organized and relevant. No major changes needed.

Re. 25: Thank you again for this positive assessment of our study and for taking the time to review it!

See document "Response_20260202".

---

## [Decision Letter · Decision Letter 1]

23 Feb 2026

Thiamine status of whitefish (*Coregonus maraena* ) in the Baltic Sea

PONE-D-25-60810R1

Dear Dr. Marc Maximilian Hauber,

We’re pleased to inform you that your manuscript has been judged scientifically suitable for publication and will be formally accepted for publication once it meets all outstanding technical requirements.

Kind regards,

Amel Mohamed El Asely

Academic Editor

PLOS One

Additional Editor Comments (optional):

Reviewers' comments:

Reviewer's Responses to Questions

**Comments to the Author**

Reviewer #1: All comments have been addressed

Reviewer #2: All comments have been addressed

2. Is the manuscript technically sound, and do the data support the conclusions?

Reviewer #1: Yes

Reviewer #2: Yes

3. Has the statistical analysis been performed appropriately and rigorously?

Reviewer #1: Yes

Reviewer #2: Yes

4. Have the authors made all data underlying the findings in their manuscript fully available?

Reviewer #1: Yes

Reviewer #2: Yes

5. Is the manuscript presented in an intelligible fashion and written in standard English?

Reviewer #1: Yes

Reviewer #2: Yes

Reviewer #1: (No Response)

Reviewer #2: Dear Editor,

I would like to thank the authors for their careful revision of the manuscript and for their detailed responses to the comments. The authors have adequately addressed the main concerns, and the manuscript has improved in clarity and quality.

In my opinion, the manuscript is suitable for publication in its current form.

Recommendation: Accept.

Sincerely,

Reviewer

**Do you want your identity to be public for this peer review?** For information about this choice, including consent withdrawal, please see our Privacy Policy

Reviewer #1: No

Reviewer #2: No

---

## [Editor Report · Acceptance letter]

PONE-D-25-60810R1

PLOS One

Dear Dr. Hauber,

I'm pleased to inform you that your manuscript has been deemed suitable for publication in PLOS One. Congratulations! Your manuscript is now being handed over to our production team.

Kind regards,

on behalf of

Prof. Amel Mohamed El Asely

Academic Editor

PLOS One